# Machine Learning-Based Prediction of the Compressive Strength of Brazilian Concretes: A Dual-Dataset Study

**DOI:** 10.3390/ma16144977

**Published:** 2023-07-13

**Authors:** Vitor Pereira Silva, Ruan de Alencar Carvalho, João Henrique da Silva Rêgo, Francisco Evangelista

**Affiliations:** Department of Civil and Environmental Engineering, SG-12, University of Brasília (UnB), Brasilia 70910-900, Brazil; runcarvalho@gmail.com (R.d.A.C.); jhenriquerego@unb.br (J.H.d.S.R.); fejr@unb.br (F.E.J.)

**Keywords:** concrete strength, machine learning, prediction, artificial neural networks, concrete, Portland cement

## Abstract

Lately, several machine learning (ML) techniques are emerging as alternative and efficient ways to predict how component properties influence the properties of the final mixture. In the area of civil engineering, recent research already uses ML techniques with conventional concrete dosages. The importance of discussing its use in the Brazilian context is inserted in an international context in which this methodology is already being applied, and it is necessary to verify the applicability of these techniques with national databases or what is created from national input data. In this research, one of these techniques, an artificial neural network (ANN), is used to determine the compressive strength of conventional Brazilian concrete at 7 and 28 days by using a database built through publications in congresses and academic works and comparing it with the reference database of Yeh. The data were organized into nine variables in which the data samples for training and test sets vary in five different cases. The eight possible input variables were: consumption of cement, blast furnace slag, pozzolana, water, additive, fine aggregate, coarse aggregate, and age. The response variable was the compressive strength of the concrete. Using international data as a training set and Brazilian data as a test set, or vice versa, did not show satisfactory results in isolation. The results showed a variation in the five scenarios; however, when using the Brazilian and the reference data sets together as test and training sets, higher R^2^ values were obtained, showing that in the union of the two databases, a good predictive model is obtained.

## 1. Introduction

Concrete has been widely used as a construction material worldwide, becoming one of the most commonly used materials in construction sites in many countries. Due to the complexity of its properties, the characteristics of fresh and hardened concrete are frequently and thoroughly studied. Concrete preparation involves the use of cement, fine and coarse aggregates, chemical admixtures, and water. When cement reacts with water, the chemical reaction of hydration occurs slowly, and the hydrated calcium silicate is formed (C-S-H). It bonds the aggregate particles with the paste. This hydration process begins immediately with the addition of water and continues for years to make concrete a solid, stable, and resistant material [1,2]

Through this chemical process, concrete acquires a property of great relevance to structural engineering: axial compressive strength. Expressed in MegaPascals (MPa), it is obtained through the uniaxial compression test on cylindrical specimens using a press standardized by the Brazilian standard NBR 5739 [3]. Every country in the world uses a specific standard with distinct test parameters. Therefore, this standard is only applicable in Brazil. This test consists of applying a uniformly distributed axial compression load at a constant speed on the base of the specimen until the material ruptures.

Some studies [4,5,6] suggest that concrete compressive strength may be directly related to the water-to-cement ratio since the higher this factor, the lower the observed strength. However, it is noted that this ratio is not the only relevant factor in compressive strength. Other elements, such as cement properties, aggregate particle size distribution, mixture proportions, chemical admixtures, and supplementary cementitious materials (such as fly ash and slag), may also influence this property [7].

In the field of civil engineering research, compressive strength has been widely accepted as the most important mechanical property when compared to Poisson’s ratio and Young’s modulus by structural engineers. As discussed, in some cases, it is necessary, for example, to wait for 28 days to obtain this parameter. This waiting time is not attractive to researchers and engineers who intend to shorten this period for the study of new chemical admixtures, for example [7].

Numerous studies have been striving to develop models capable of predicting the compressive strength of materials such as cement, mortar, and concrete. These models use different methodologies and data sources, such as statistical techniques, analytical analyses, mathematical calculations, numerical simulations, and computational algorithms [8,9,10,11,12].

Researchers [13,14,15] have started to use machine learning (ML) techniques to estimate the compressive strength of concrete, showing its high problem-solving capabilities in the construction industry, including quality management, project optimization, fracture mechanics, structural monitoring, and risk analysis.

ML, which is a subfield of artificial intelligence (AI), works with algorithms that enable computers to learn from a specific database [16]. It is worth noting that many studies have been carried out to predict compressive strength by using ML techniques on special concretes such as those with ultra-high strength. These require more rigorous material dosage studies, leading to complex nonlinear relationships [17]. Additionally, these models are established by dividing input data into training, testing, and validation sets. During the training phase, the model autonomously learns the main patterns of the data. The validation set helps to confirm the predicted results during training and eventually during the testing step or when new data not included in the training set are introduced. This step is used to evaluate the performance of the developed model [18].

The artificial neural networks (ANN) model is among the most commonly used to predict compressive strength and was the model used in this research [19]. To predict the compressive strength of conventional and unconventional concretes, the ANN technique works similarly to the human brain, which is widely known in the computational field [20,21,22]. The first layer of the algorithm is the input data, where each one is filled and generates an output data return. This technique is highly used for strength prediction studies because it presents versatility and simplicity without requiring the knowledge of more complex programming languages.

Different factors that influence concrete compressive strength are used as input variables in these studies [16,23]. Xu et al. [24] identified variables, such as the water/cement ratio, maximum aggregate size, and aggregate/cement ratio in a study on concrete incorporating recycled aggregate, for example. Furthermore, Yeh [17] investigated the potential use of experiments and modeling with neural networks to determine the effect of fly ash substitutions, from 0 to 50%, on the late compressive strength, from 3 to 56 days, of low- and high-strength concretes. The study concluded that neural networks, after training, can help verify which components contribute most to obtaining higher strength and suggests that mixtures with higher contents produce lower strength ratios in the late period.

Machine learning techniques, particularly the integration of gray box modeling, have garnered significant scholarly attention in the realm of predicting various properties, including shear capacity, in reinforced concrete structures [25]. This novel approach combines mechanics-based formulations with machine learning methodologies, thereby enhancing the precision and robustness of predictive models while preserving the underlying physical understanding of the resisting mechanisms. Additionally, the utilization of machine learning algorithms facilitates the prediction of additional concrete properties, such as flexural strength and durability, by effectively capturing intricate patterns and correlations within extensive datasets [24,26,27]. Consequently, the integration of machine learning with traditional engineering knowledge presents a promising avenue for advancing the field of concrete engineering, enabling the design and optimization of more sustainable and efficient structures [28,29].

This article is justified in a context where the ML approach for predicting compressive strength in concretes manufactured in Brazil is still low, and there is a trend that it will be increasingly investigated for different contexts in civil construction. Thus, the main objective of this study is to compare the results of five ML models developed using two datasets: a national database, constructed through publications in congresses and academic works; and a reference database. These models are built using multilayer perceptron (MLP), a type of artificial neural network (ANN).

## 2. Materials and Methods

The research was conducted following the steps outlined below:

### 2.1. Step 1—Obtaining the Brazilian Dataset

The study was based on two distinct databases. The first was the Yeh [17] database, which is freely available on the UC Irvine Machine Learning Repository and contains 1030 instances for analyzing different ML algorithms. These data were computed and analyzed using statistical and numerical parameters. The input data included cement (kg/m^3^), blast furnace slag (kg/m^3^), pozzolana (kg/m^3^), water (kg/m^3^), coarse aggregate (kg/m^3^), fine aggregate (kg/m^3^), and age (days), with the output being predicted compressive strength (MPa). These input data were chosen because one of the main objectives of the analysis was to determine how dosage parameters affect strength. This dataset will be denominated as DATA_YEH98.

The second database was constructed by the authors and was based on the same input variables but for Brazilian concretes, obtained from the literature, focusing on theses, dissertations, and national articles (DATA_BR23). In total, 324 distinct instances were obtained for analysis, with over 70% taken from articles published in the IBRACON congresses in 2018, 2019, and 2020. Established in 1972, this Brazilian organization comprises professionals and industry participants involved in the production of concrete. Its primary mission is to promote scientific and technological research related to concrete, whether it be as a material or structure, and to advance the best practices within the construction sector.

The data available in the repository are divided into 7 components, all in kg/m^3^: cement, slag, fly ash, water, superplasticizer, coarse aggregate, and fine aggregate. However, Brazilian concrete dosage data are usually presented in a unitary form, such as 1:2:3, indicating one-part cement to two-part sand and three parts gravel. To estimate the amount of each component in the concrete, the cement consumption was used as a reference, along with an average estimate of supplementary cementitious materials, since these components vary according to the type of cement used.

An adjustment was necessary, as Brazilian cement may have specific limits for clinker with calcium sulfate (equivalent to the input data for cement), slag, pozzolana, and filler. To ensure that the analysis parameters were the same as those in the repository database, compatibility was achieved according to the year of publication and the Brazilian standard in effect at that time. The “Fly Ash” input was replaced by “Pozzolana” since the Brazilian standards use this more general term for estimating cement compositions. The NBR 16697 [30] standard was used from 2018 onwards, while the NBR 11578, NBR 5735, and NBR 5733 were used in previous periods. Table 1 shows the percentage values considered after and before 2018, respectively.

The data for each of the above components were estimated based on the simple arithmetic mean. Step 1 corresponds to the data mining stage obtained for the national database. Data mining is the process of extracting useful information and significant patterns from large datasets. In summary, this is the essential step for constructing the database that will be modeled later.

To determine which inputs would be considered in the ANN model, Pearson’s correlation was used [31]. It is a statistical measure that evaluates the linear relationship between two variables. It can range from −1 to 1, with the lower limit indicating a perfect negative correlation and the upper limit indicating a perfect correlation. Finally, 0 corresponds to no correlation. A Pearson correlation matrix was applied to assess the linear relationship among the independent variables and detect multicollinearity [32] and also to evaluate if all the input variables have some correlation with the response variable.

### 2.2. Step 2—Training Using Artificial Neural Networks (ANN)

A standard process for training, validation, and testing artificial neural networks involves specific steps. The first step is data preparation, which involves organizing and adjusting the data for use in the neural network. This includes activities such as normalization, the imputation of missing values, and the separation of data into training and testing sets [9].

Next, the neural network training step is performed, where the training set is used to adjust the weights and layers of the network so that the predicted outputs are close to the actual values. Subsequently, the network testing step is executed, in which the testing set is used to evaluate the accuracy of the trained neural network. This is achieved by comparing the predicted outputs with the actual values of the test data [13].

Based on the use of some polynomials [33], it can be argued that models should strike a balance between underfitting and overfitting situations. In the case of overfitting, the model fits excessively well with the training data but fails to exhibit good generalization. On the other hand, underfitting occurs when the model is too simplistic for the dataset, consequently failing to capture the underlying patterns and exhibiting poor generalization.

To avoid the aforementioned situations, it is crucial to measure the prediction errors for both the training data and unseen data, thereby engaging in a process called validation. In this regard, cross-validation techniques come into play, utilizing the training data itself for validation purposes. By employing this approach, there is no reduction in the training set, and there is no need to acquire additional data solely for validation. These aspects hold significant value as data acquisition is an expensive endeavor, demanding time, effort, financial resources, organization, and expertise [33].

Furthermore, it is important to emphasize that even for simple machine learning problems [34], thousands of examples are required. Thus, the use of cross-validation techniques, which enable the accurate evaluation of model performance without the need for additional data, proves highly advantageous in the field of construction and other domains facing challenges related to data collection. In this research, the k-fold cross-validation with 10 folders was used.

If the network accuracy is not satisfactory, adjustments to the network architecture or training parameters can be made, and the training and testing process can be repeated. Finally, when the accuracy is satisfactory, the neural network can be used to make predictions with new data. The ultimate goal of this process is to adjust the neural network so that it can make accurate predictions with new and unknown data.

Many predictive pieces of training use linear regression (LR), a supervised machine learning algorithm that uses known historical data to predict behaviors. This model is widely applicable, which justifies its frequent use and establishes a relationship between the dependent variable and the independent variable.

For an analysis of the training and testing data, in this article, we opted for the methodology of dividing the data into 5 main cases, as simplified in Table 2 and Figure 1.

In Cases 1, 2 and 5, 80% of the samples were used for training and 20% for tests, while in Cases 3 and 4, it was employed in 100% of the samples.

The training was conducted using the Python programming language, widely employed in machine learning applications. Figure 2 shows the structure of a type of machine learning model, the multilayer perceptron, a typical ANN architecture.

The regression was conducted using the multilayer perceptron (MLP) architecture which is a neural network with one or more hidden layers with an indeterminate number of neurons. The MLP is a fully connected class of feedforward neural networks, which implies that all neurons in each layer are connected to all neurons in the subsequent layer, allowing the input signal to propagate only in a forward direction toward the output.

This technique is commonly used when obtaining a backpropagation algorithm with a neural output layer. The analysis involves adjusting the hyperparameters used for better model fitting. It, in a nutshell, applies gradient descent (GD), also known as the method of the gradient, along with an effective technique for calculating gradients automatically. For each training instance, initial predictions are made while preserving the intermediate results, and the error is measured by comparing the predicted values with observed values. Then, by back-propagating through the layers of neurons until reaching the input layer, it becomes possible to measure the contribution of each connection to the error. Finally, with measurements of the error gradient with respect to the weight of each connection, applying GD allows for adjusting all the network’s weights and reducing the errors.

There is no specific prediction formula for modeling. In this research, when using MLP, the technique can be called a “black box,” meaning that although it is capable of learning and making decisions based on input data, studying its structure will not give the function that is being approximated by the model. Since the analysis of weights for each connection from one layer to the next may not provide accurate enough information about feature importance, the permutation importance [35] of the models with the most accuracy on the test set was calculated.

Within the scope of the study, the permutation importance technique was employed to assess the relative importance of the input variables in an artificial neural network (ANN) model. By permuting the values of each input variable while keeping others unchanged and then comparing the model’s performance metrics before and after permutation, the technique identified the variables that had the most impact on the model’s predictions. This approach helped in identifying the most relevant variables and understanding variable relationships [36].

### 2.3. Step 3—Statistical Analysis of the Technique according to the Presented Cases

There are three main statistical criteria applied to evaluate the error between the observed and predicted values of concrete compressive strength for the ANN training methodology. To measure the model’s performance, the coefficient of correlation (R^2^), mean squared error (MSE), and root mean squared error (RMSE), calculated according to Equations (1), (2), and (3), respectively, was employed.

The R^2^ is used to assess the linear correlation between the observed and predicted values in the range. It measures the model’s ability to explain variations in the result, indicating how predictive the model is. Its value can range from 0, indicating no variability in the result (random chance), to 1, indicating that the model explains all the variabilities (total accuracy).
(1)$$R^{2}=1-\frac{\sum_{n-1}^{n}{\left({\hat{y}}_{i}-y_{i}\right)}^{2}}{\sum_{n=1}^{n}{{(y}_{i}-\overset{-}{y})}^{2}}$$

The mean squared error (MSE) measures the average squared error of the model predictions, calculating the difference between the experimental result and the predicted values and then calculating the mean. Finally, the root mean squared error (RMSE) measures the difference between the predicted values and the actual squared values inserted into a root. They can be equated as a “standard deviation of errors”.
(2)$$\mathrm{MSE}=\frac{1}{n}\sum_{i=1}^{n}{\left({\hat{y}}_{i}-y_{i}\right)}^{2}$$
(3)$$\mathrm{RMSE}=\sqrt{\frac{1}{n}\sum_{i=1}^{n}{\left({\hat{y}}_{i}-y_{i}\right)}^{2}}$$

These metrics enable the evaluation of the model’s quality with good accuracy, allowing for a comparison between different models and cases to be analyzed. For the MSE and RMSE indicators, a lower value is desirable, as it represents a better model, whereas, for R^2^, a higher value represents a closer fit to the ideal and more likely prediction.

## 3. Results

In this topic, we present and discuss the results of the three stages of the research. Firstly, we describe the development of the Brazilian dataset based on those made available in the repository. Next, we train a model for each of the five cases, which differs according to the data used for training and testing. Finally, we provide a detailed analysis of the two best cases.

### 3.1. Step 1—Obtaining the Aligned Brazilian Dataset

In this stage, data selection, pre-processing, transformation, mining, analysis, and results assimilation were performed. Case 1 was obtained after training and testing solely on the Brazilian dataset (DATA_BR23), using a proportion of 20–80% for each set.

The UC Irvine Machine Learning Repository dataset was used in Yeh’s study [13], which serves as a reference for many concrete strength prediction research studies. In this case, Yeh used 1080 instances to demonstrate the adaptability of ANNs in predicting this property in high-performance concrete. A set of concrete mixes was produced in the laboratory, and two main conclusions were drawn: an ANN-based strength model is more accurate than a linear regression-based model [9,19], and it is convenient and easy to use ANN models for numerical experiments to review the effects of each variable proportion in the concrete mix. This was the dataset demonstrated for Case 2 in this paper, separating the sample in 20–80% proportions for comparison with Case 1.

The construction of the Brazilian dataset began after analyzing the conclusions from training using only the repository dataset. To align the Brazilian dataset with the repository dataset and quantify the cementitious materials, input data were obtained from the first three instances. The data were organized in table form, with each column representing a component.

The complete Brazilian dataset is exemplified in Table 3.

When analyzing the two databases, it is important to evaluate Pearson’s correlation between each variable, as shown in Figure 3 and Figure 4, so that it helps with feature selection. Concerning the target variable, cement was the input with the highest Pearson’s correlation, an absolute value close to 1, with values of 0.5 for YEH98 and 0.51 for BR2023, indicating a strong relationship with compressive strength and thus justifying its selection. Cement is the primary component of concrete, with binding properties directly related to its hardened state characteristics.

Regarding the correlation between inputs, the correlation between water and admixture stands out with a correlation of −0.66 in YEH98. In neither of the databases did we see a correlation higher than 0.9 among the inputs, which is a positive aspect because if it were to occur, it would lead to multicollinearity, thus potentially interfering with the model.

It can also be highlighted through correlation matrices that, in both databases, the input pozzolan shows the lowest correlation with compressive strength. For this reason, it was decided not to select this variable, being the only one from the dataset not used in the models.

This can be justified, primarily because when estimating the quantities of pozzolana and blast furnace slag in BR2023, based on the average cement consumption prescribed by the Brazilian standards, the calculation was not sufficient to reflect the actual amount of material in the cement samples collected from the literature, unlike YEH98, where the inputs were relevant. Considering the knowledge in the materials field, it is known that these supplementary cementitious materials directly influence compressive strength. Thus, all other inputs, except for pozzolana, were used in the models.

### 3.2. Step 2—Training of the ML Technique Using ANN

Below is Table 4 with the maximum (Max), average (μ), minimum (Min), and standard deviation (σ) predicted strength values obtained for each case:

In comparison to Case 2, Case 1 (TR_TE.BR) shows a low variation in the mean between the training and testing samples, with a standard deviation of 10.99 and 11.50, respectively. This can be explained by the fact that, in Case 2, the database used data from various countries, whereas the Brazilian one only utilized national data.

However, in Cases 3 and 4, more significant differences can be observed between the same dataset used for training and testing. In these two cases, different datasets were swapped between training and testing. The difference between the mean values of the DATA_YEH1998 dataset was 12.34%, while for DATA_BR2023, it was 9.24%. In this case, the use of different databases in training and test sets may have been responsible for a higher variation.

Thus, in all the presented cases, the predicted mean values were higher than 25 MPa and lower than 40 MPa, indicating that the analyzed concretes belong to Class I, which comprises concretes with a strength between 25 MPa and 50 MPa. Case 5 showed the highest variance, with an average amplitude of 69.59 MPa and σ = 16.06 in the test results.

### 3.3. Step 3—Statistical Analysis of the Technique according to the Presented Cases

Table 5 summarizes the results of the statistical parameters for each case.

It can be observed that the best results for training were, in ascending order: Case 4, Case 3, Case 1, Case 5, and Case 2. For testing, also in ascending order: Case 3, Case 4, Case 1, Case 5, and Case 2.

Cases 1, 2, and 5 were satisfactory for modeling, as they had a more linear fit and behavior. On the other hand, Cases 3 and 4 varied in a way that made the fit more difficult. Thus, using Brazilian data to train a model and testing it with data from the repository is not suitable, and vice versa.

Concrete is a material with nonlinear characteristics, and explaining its behavior is a complex task. In this sense, using an ANN is a specific and justified choice, according to authors [13,14,15,16,17,18,19,20,21,22,23], and depends on the dataset used. In Yeh’s study [13], using four distinct models with varying inputs, RMSE values between 2 MPa and 4.5 MPa were obtained with the augment-neuron networks for both testing and training. This parameter assesses the accuracy of the network, and thus, in Cases 3 and 4, there is not much accuracy.

In another study, Yeh [17] used an ANN in the same case to predict the strength of high- and low-strength concretes based on the variation of fly ash, obtaining, in the best case, an RMSE of 3.96 MPa (R^2^ = 0.890) and 8.82 MPa (R^2^ = 0.791) for training and testing data, respectively. Thus, only in the testing of Case 3 was an RMSE of 9 MPa (R^2^ = 0.43) obtained, once again reinforcing that the model did not have good performance.

Therefore, the best for both training and testing was Case 2, in which the RMSE presented values of only 1.83 and 4.20, respectively, with only the training measure being acceptable in terms of statistical errors. This means that, in this first case, the model may be wrong by more or less 1.83 MPa while obtaining an R^2^ of 0.99.

The results of the histograms for (a) DATA_BR2023 and (b) DATA_YEH98 in the measurement units (on the y-axis) of Table 3, for each input, and the frequency on the x-axis. for each parameter can be observed in Figure 5 and Figure 6.

Through the histograms of the input data, a wide range of data can be observed, which were collected only at the ages of 7 and 28 days. The blast furnace and pozzolan exhibited similar behavior, likely due to being quantified approximately, as well as the utilization of various types of cement. Similarly, the frequency curves of fine aggregate and water also exhibit some resemblance.

In the DATA_YEH98 dataset, considering the reference database, a greater variety was observed in terms of the days on which the specimen was ruptured for the test, with the majority concentrated within 50 days. The achieved compressive strengths, reaching approximately 200 MPa, are twice as high as the highest values analyzed in the DATA_BR2023 dataset.

The results of the graphs with the experimental and predicted resistance can be observed in each of the cases shown in Figure 7.

An equality line is plotted, represented by the dashed line in both graphs. It is important to note that the predicted values are close to the linear fit, confirming an agreement with the concrete compressive strength values, especially in Case 2, using only the DATA_YEH1998 dataset, and in Case 5, using all databases.

The charts in Figure 4 demonstrate that Case 2 is the best fit when compared to the others; therefore, the model analysis concludes that it is the ideal one. However, this case only considers the Yeh (1998) database. Case 5 presents the second-best performance and considers the DATA_BR2023 and DATA_YEH98 samples as training, and the test presents an RMSE value of 2.62 in training and 6.01 in testing.

It is worth noting that Case 4, during the adjustments, was subdivided into three others to improve the fitting, justified by the variation in the cases, as 100% of the samples were used. To achieve this, the hyperparameters were manually modified: the number of neurons in the layer and the alpha factor for a more underfitting direction to better generalize the test samples. The smaller the alpha, the more overfitting behavior, and the larger, the more underfitting. These modifications justify the differences in the results of the R^2^, MSE, and RMSE parameters, as well as the differences in the samples between the cases.

Finally, the permutation importance analysis was performed using Cases 2 and 5, as they showed the best overall results. A total of 500 permutations were conducted for each input variable individually, and this tool measures how much the RMSE decreases, allowing us to understand which variables were more relevant for the models. It is demonstrated in Table 6 and Figure 8.

In Case 2, the order of relevance, in ascending order, was: age, cement, water, fine aggregate blast furnace slag, coarse aggregate, and admixture. In Case 5, age and cement remained the two most relevant variables; however, in a different order, as illustrated in Figure 8, it was in the following sequence: cement, age, blast furnace, water, admixture, fine aggregate, and coarse aggregate.

## 4. Conclusions

This study investigates the relationships between the conventional concrete constituents and the corresponding compressive strength using a multilayer perceptron (MLP) algorithm. A dataset containing 324 experiments on Brazilian concretes was used to generate the training and testing datasets for developing the MLP algorithm. The considered concrete components (inputs) were the following eight: cement (kg/m^3^), blast furnace slag (kg/m^3^), water (kg/m^3^), coarse aggregate (kg/m^3^), admixture (kg/m^3^), fine aggregate (kg/m^3^), and age (days). The output was the compressive strength, and the main goal was to compare two datasets using three well-known statistical measurements, such as RMSE, MSE, and R^2^.

It is concluded that the best artificial neural network (ANN) models were the ones in Case 2, using only the repository database, and in Case 5, when adding the Brazilian concrete database together. Both cases presented satisfactory statistical measures and parameters that could still be improved with other machine learning models or new configurations using new databases.

Furthermore, in Case 1, using only the Brazilian database, good training performance was observed, but it may be necessary to increase the number of instances with more inputs that can better explain the observed variations. In Case 4, the worst results were observed when using the Brazilian database for training. An analysis of the variance between the results and standard deviation shows that the parameters vary significantly, even with the same input data.

In the validation step, the quality of the models dropped sharply, with the best R^2^ testing occurring only with the Brazilian dataset being 0.77. The probable main contribution to this result was that, in DATA_BR2023, although the values were collected within the same country, the variations were relatively large when compared to the sample size. Thus, when comparing it to the database in the repository (represented by Case 2), standard deviations of similar magnitudes can be observed.

The Brazilian sample does not generalize well to the repository dataset and vice versa. The ideal for modeling proved to be either using the databases separately for training and testing in each case or using both together, as shown in Case 5. The best two inputs related to compressive strength in Cases 2 and 5 were age and cement.

The accuracy of prediction models in concrete engineering is notably affected by various factors, including the completeness, quantity, quality, and distribution of input parameters within the dataset. These aspects play a critical role in determining the reliability and robustness of the models. Therefore, it is essential for the authors to thoroughly discuss these factors when presenting their research findings.

When considering why a dataset may exhibit greater uniformity compared to another, even when using data from the same country, several factors come into play. Firstly, the selection criteria for data collection can contribute to varying degrees of uniformity. If the data collection process primarily targets specific regions, construction practices, or periods, it may result in a biased dataset with a limited representation of the overall population. Secondly, the availability and accessibility of data sources can differ across regions or construction projects, leading to varying levels of data completeness. Additionally, the quality of data can be influenced by factors such as the accuracy of measurement techniques, consistency in data recording practices, and the presence of outliers or missing values.

As a recommendation for future studies, it is advisable to enhance the dataset and expand the data volume. Additionally, it is essential to incorporate additional factors on the compressive strength of concrete, thereby obtaining more comprehensive data to facilitate accurate predictions of compressive strength for special types of concrete, such as recycled concrete and steel-fiber concrete. For other types of mixtures and specific concretes, the same methodology can be comparatively employed in different cases, with the analysis of inputs and outputs to be considered.

This study can assist other researchers in understanding how the analysis of two databases obtained from different contexts can be utilized for training and testing neural network models. The conducted permutation importance analysis, for instance, identifies the meaningful inputs for each database, influencing decision-making in subsequent research and aiding feature selections for other researchers. However, it should be noted that the DATA.BR2023 database, unlike the DATA.YEH98 database, will not be publicly available for consultation, and the programming codes used for model development are also not provided. For further details, readers are advised to consult the authors.

Finally, this finding illustrates that the regionalization and homogeneity of certain datasets can result in false-positive outcomes when searching for universally applicable concrete mix design strategies. In a future investigation, the authors aim to quantitatively evaluate the generalizability of models. Additionally, collaborative efforts are necessary to construct a comprehensive and diverse database encompassing various concrete properties. Until such a database is available, the authors recommend that machine learning models for concrete mix design be limited to predicting the strength of specimens from the same laboratories where they were trained. In summary, this study demonstrates the potential viability of machine learning techniques in predicting concrete compressive strength. Nonetheless, further research is required to establish larger and more diverse databases, which may ultimately reduce the time and resources currently expended in the mix design processes.

## Figures and Tables

**Figure 1 materials-16-04977-f001:**
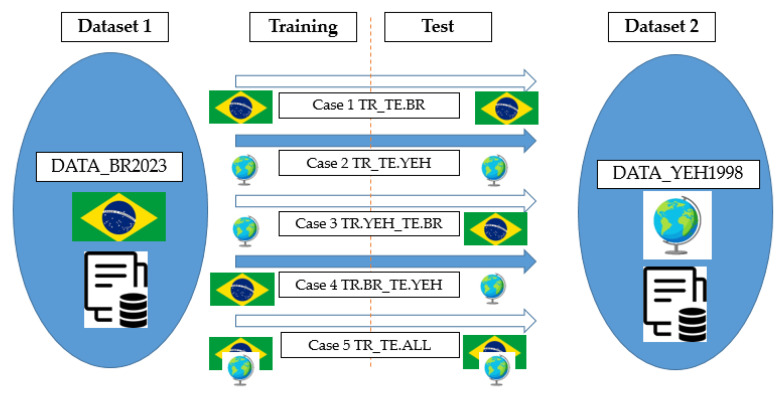
Visual representation of the methodology for case division.

**Figure 2 materials-16-04977-f002:**
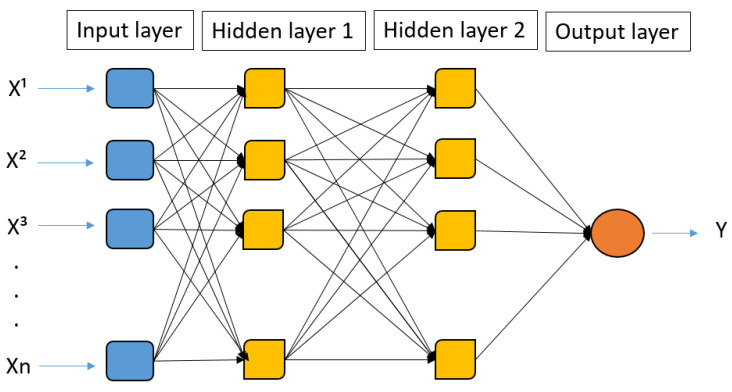
Multilayer perceptron architecture.

**Figure 3 materials-16-04977-f003:**
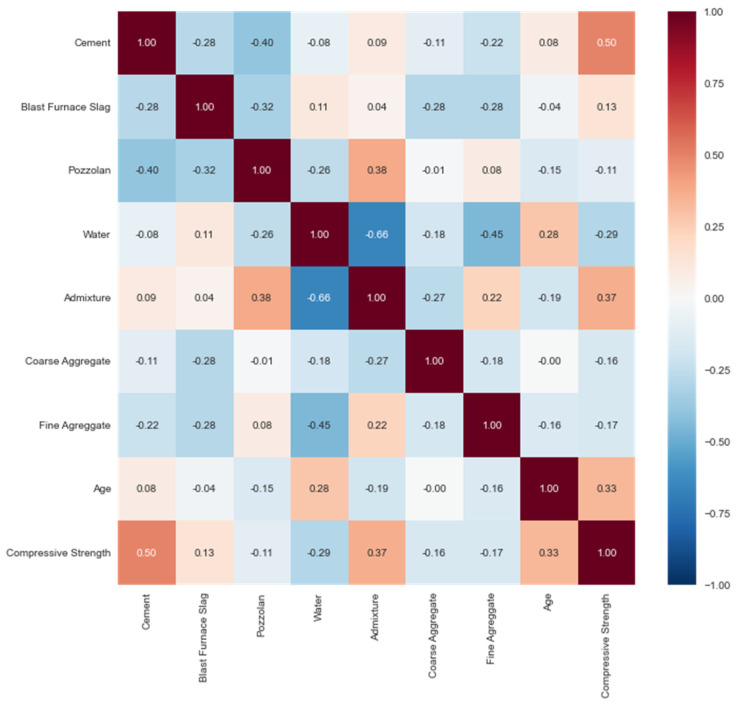
Pearson’s correlation in YEH98.

**Figure 4 materials-16-04977-f004:**
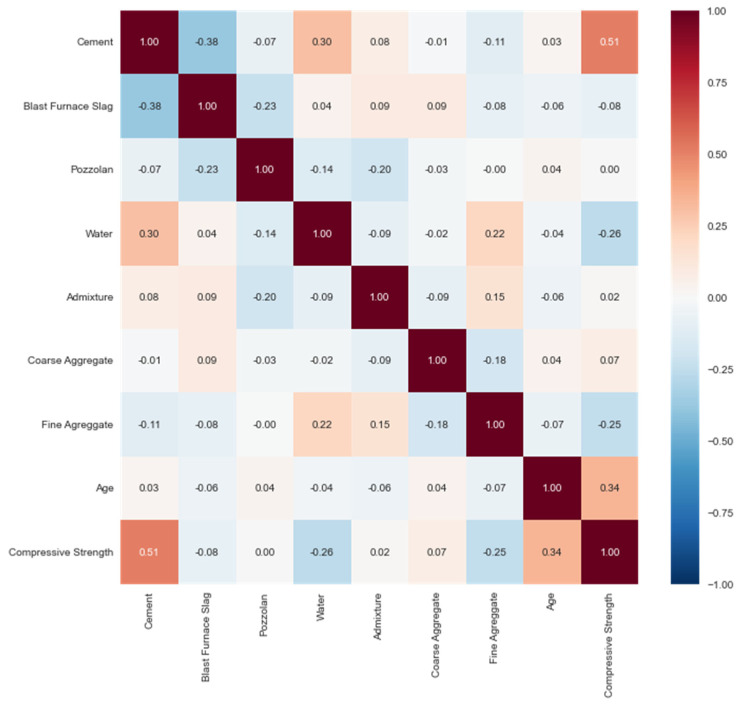
Pearson’s correlation in BR2023.

**Figure 5 materials-16-04977-f005:**
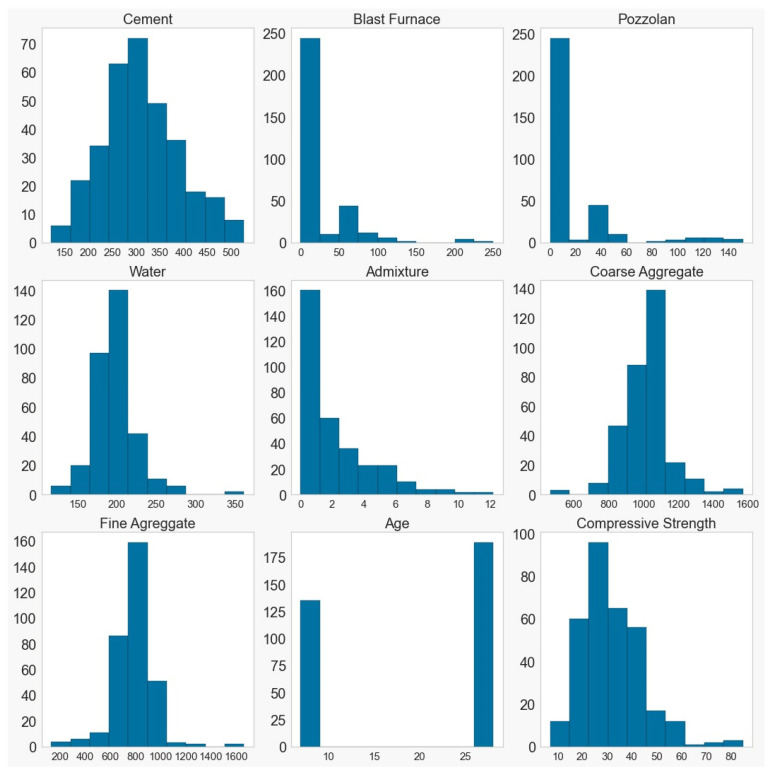
Histograms of the inputs in the Brazilian database.

**Figure 6 materials-16-04977-f006:**
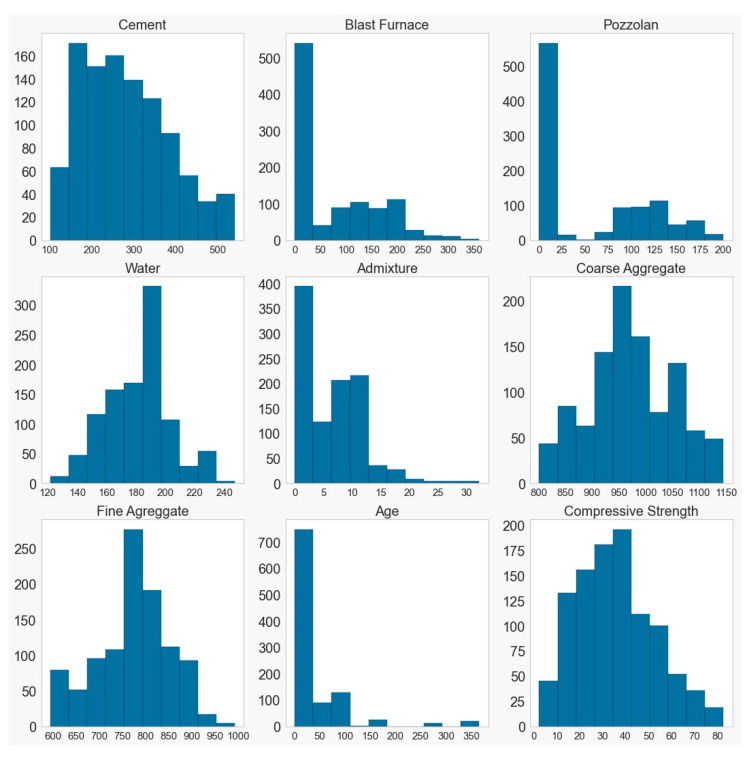
Histograms of the inputs in the repository database.

**Figure 7 materials-16-04977-f007:**
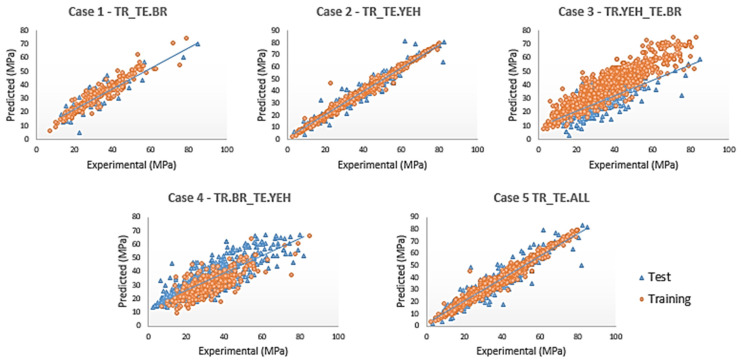
Regression charts for each case using artificial neural networks (ANN).

**Figure 8 materials-16-04977-f008:**
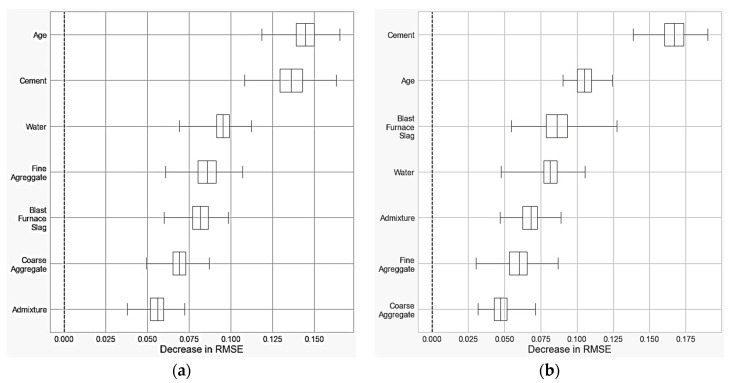
Permutation importance analysis in Case 2 (**a**) and Case 5 (**b**).

**Table 1 materials-16-04977-t001:** Average values adopted in the Portland cement’s composition.

Average Values Adopted—NBR 16697 (After 2018/Before)					
	Clinquer + Calcium Sulfate	Slag	Pozzolana	Fíler	Total
CP II E	72.5/75%	20.0%	0.0%	7.5/5.0%	100.0%
CP II Z	82.5/85%	0.0%	10.0%	7.5/5.0%	100.0%
CP II F	82.0/92%	0.0%	0.0%	18.0/8.0%	100.0%
CP III	45.0%	50.0/52.5%	0.0%	5.0/2.5%	100.0%
CP IV	65.0%	0.0%	30.0/30.2%	5.0/2.5%	100.0%
CP V ARI	90.0/ 97.5%	0.0%	0.0%	10.0/2.5%	100.0%

**Table 2 materials-16-04977-t002:** Division of cases for predictions.

Case	Training	Test	Acronym
1	DATA_BR23	DATA_BR23	TR_TE.BR
2	DATA_YEH98	DATA_YEH98	TR_TE.YEH
3	DATA_YEH98	DATA_BR23	TR.YEH_TE.BR
4	DATA_BR23	DATA_YEH98	TR.BR_TE.YEH
5	ALL	ALL	TR_TE.ALL

**Table 3 materials-16-04977-t003:** Example of DATA_BR2023.

DATA		Cement (kg/m^3^)	Blast Furnace Slag (kg/m^3^)	Pozzolan (kg/m/^3^)	Water (kg/m^3^)	Admixture (kg/m^3^)	Coarse Aggregate (kg/m^3^)	Fine Aggregate (kg/m^3^)	Age (days)	Compressive Strength (MPa)
BR2023	Máx	526.50	250.00	151.31	361.10	12.21	1573.00	1662.29	28.00	84.90
	μ	310.22	20.31	14.75	197.39	1.97	1016.83	786.94	19.30	32.13
	Min	121.50	0.00	0.00	116.33	0.00	463.13	127.00	7.00	7.00
	σ	80.01	41.43	3.49	27.82	2.42	138.16	154.11	10.36	12.27
YEH98	Máx	540.00	359.40	200.10	247.00	32.20	1145.00	992.60	365.00	82.60
	μ	281.17	73.90	54.19	181.57	6.20	972.92	773.58	45.66	35.82
	Min	102.00	0.00	0.00	121.75	0.00	801.00	594.00	1.00	2.33
	σ	104.51	86.28	64.00	21.36	5.97	77.75	80.18	63.17	16.71

**Table 4 materials-16-04977-t004:** Predicted strength (MPa).

Predicted Strength (MPa)			
		Training	Test
Case 1 TR_TE.BR	Máx	74.55	70.62
	μ	31.95	31.83
	Min	6.2	5.13
	σ	10.99	11.50
Case 2 TR_TE.YEH	Máx	79.07	81.43
	μ	35.73	36.21
	Min	6.44	4.43
	σ	16.78	16.62
Case 3 TR.YEH_TE.BR	Máx	75.19	59.47
	μ	38.06	28
	Min	8.92	2.71
	σ	15.97	9.43
Case 4 TR.BR_TE.YEH	Máx	66.83	67.37
	μ	30.85	33.36
	Min	13	13.53
	σ	9.237	11.63
Case 5 TR_TE.ALL	Máx	78.7	83.25
	μ	34.83	35.08
	Min	9.11	2.27
	σ	11.69	16.06

**Table 5 materials-16-04977-t005:** Statistical parameters.

Statistical Parameters			
		Training	Test
Case 1 TR_TE.BR	R^2^	0.9	0.77
	MSE	14.12	40.63
	RMSE	3.76	6.37
Case 2 TR_TE.YEH	R^2^	0.99	0.94
	MSE	3.35	17.67
	RMSE	1.83	4.2
Case 3 TR.YEH_TE.BR	R^2^	0.8	0.93
	MSE	56.2	86.33
	RMSE	7.5	9.29
Case 4 TR.BR_TE.YEH	R^2^	0.6	0.67
	MSE	60.64	77.56
	RMSE	7.78	8.81
Case 5 TR_TE.ALL	R^2^	0.97	0.86
	MSE	6.88	36.11
	RMSE	2.62	6.01

**Table 6 materials-16-04977-t006:** Permutation importance.

Input	Permutation Importance			
	Case 2		Case 5	
	RMSE (μ)	σ	RMSE (μ)	σ
Cement	0.136	0.01	0.167	0.009
Age	0.144	0.008	0.105	0.007
Blast Furnace Slag	0.082	0.007	0.086	0.011
Water	0.095	0.006	0.082	0.007
Admixture	0.056	0.006	0.068	0.007
Fine Agreggate	0.086	0.008	0.06	0.009
Coarse Aggregate	0.069	0.006	0.047	0.007

## Data Availability

The reference database (DATA_YEH98) used in this study is available in the public repository UCI Irvine Machine Learning Repository, through the link: https://archive.ics.uci.edu/dataset/165/concrete+compressive+strength. The data created for this study (DATA_BR23) are available upon request to the corresponding author.

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
