# Peer review of "Machine Learning-Based Prediction of the Compressive Strength of Brazilian Concretes: A Dual-Dataset Study"

_materials, 2023, doi:10.3390/ma16144977_

Round 1
Reviewer 1 Report
​This study aims at predicting compressive strength of concrete by ANN based on two different databases. The capability of the ML-based model which is trained by two significantly different databases is analyzed and discussed. The following comment should be addressed by the authors:1. Some manuscript guidelines are not removed (e.g., pages 13 and 14).
2. ANN method should be described​​ more appropriately by presenting a schematic illustration and the equation. The following references might be useful:
(a) Dabiri, H., Faramarzi, A., Dall’Asta, A., Tondi, E., & Micozzi, F. (2022). A machine learning-based analysis for predicting fragility curve parameters of buildings. Journal of Building Engineering, 62, 105367.
(b) Mitropoulou, C. C., & Papadrakakis, M. (2011). Developing fragility curves based on neural network IDA predictions. Engineering Structures, 33(12), 3409-3421.
3. The correlation between inputs and output should be discussed more precisely. Pearson correlation coefficients reflect which parameters affect the output significantly. The authors are recommended to provide Pearson coefficients and discuss the results.
4. Quality of Fig. 4 should be enhanced. The legends are not readable and the scatter diagrams are ​​​blurry.
5. The authors name in Reference 3 are not given. ​​Author Response
- The manuscrift guidelines were removed.
- ANN method was described​​ with more details and a flowchart was added for better ilustration;
- Pearson correlation coefficients were added and discussed the influence in the input's choice.
- Fig. 4 (now represented by Fig. 7) were remake for better visualisation;
- Reference 3 was corrected;
Reviewer 2 Report
1. The Section 1 mainly introduces the material of concrete and the compressive strength of concrete, but the research progress of concrete compressive strength is too little, and the research results of machine learning in the prediction of concrete compressive strength are not enough, so this content should be supplemented, and tables can be listed if necessary.
2. The two data sets of this study are presented in Section 2.1. It can be seen that the standards of Brazil concrete have certain regional characteristics, and relevant standards should be listed here.
3. In the second section of this paper, the neural network algorithm is introduced in Section 2.2, but the introduction is relatively fuzzy. The principle of the neural network algorithm is introduced, and the introduction is relatively independent. The application of the specific method in this paper is not given. It is recommended to give the schematic diagram and explain the research in detail for each layer.
4. In paragraph 9 of Section 2.2, it is introduced that the training data is 80%, and 20% of the samples are used for testing. The explanation of data division is not necessary for this paper and can be deleted. It can be combined with the next paragraph.
5. The Section 3 is titled "results," but it involves a large number of research steps. It can be divided into two parts, and the steps and results are listed separately as one part.
6. Section 3.1 describes the steps to obtain the database, but does not give the selected parameters that affect the compressive strength of concrete, which should be given here.
7. In Section 3.1, it is concluded that "the strength model based on artificial neural network is more accurate than the model based on linear regression", and specific basis should be given.
8. The steps in Section 3 can be listed as a flowchart to make it more intuitive.
9. In the results given in Section 3.2, there is no weight analysis on the influencing factors of concrete compressive strength, which should be supplemented.
10. Section 3.3 should put each model into a diagram to show the error, which is more intuitive.
11. Figure 4 of this paper is very vague, please replace it.
12. The research part of this paper has neither the prediction formula nor the weight analysis of the influencing factors. It only lists the research results and explains them. The demonstration of the model is not convincing enough.
13. The conclusion part of this paper is more inclined to the algorithm part, and should supplement the influence of this study on the compressive strength of concrete.
Author Response
-
New information regarding the input data related to compressive strength in concrete has been added, providing further references to the relationship between them and the conducted data analysis.
-
The parameters of Brazilian concretes were described in Section 2.1 and may include clinker, calcium sulfate, slag, pozzolan, and filler, as described in Table 1.
-
For a better illustration of the utilized method, Figure 2 has been included, along with two additional paragraphs in Section 2.1.
-
It was decided to retain this information as it is the only way to explain why the three statistical parameters differ in different cases.
-
The idea of repeating the subsection titles in Section 3 was precisely to clarify the theory in the methodology and the analyses in the results.
-
Pearson correlation was performed to provide parameters that affect compressive strength.
-
In Section 3.1, the sentence "the strength model based on artificial neural network is more accurate than the model based on linear regression" was referenced.
-
A flowchart outlining the basic steps was included in the methodology.
-
The results in Section 3.2 were detailed through a weight analysis of the factors influencing compressive strength in each of the databases using Pearson correlation.
-
In Section 3.3, the error diagram can be found in Figure 8, and Permutation Importance was performed.
-
Figure 4 has been redesigned and is now clearer.
-
As the model does not have a specific prediction formula, paragraphs were added explaining the concept of "black box" and the Permutation Importance analysis was conducted to demonstrate the feasibility of the model.
-
The conclusion has been supplemented with additional information on the study of compressive strength in concrete and suggestions for future research.